# LLM-as-a-Judge for assessing maintainability. First results with a causal approach.

Julien Siebert[1][0000−0002−7696−0046]

Fraunhofer Institute for Experimental Software Engineering (IESE), Fraunhofer-Platz 1, 67663 Kaiserslautern, Germany `julien.siebert@iese.fraunhofer.de`
https://www.iese.fraunhofer.de/

**Abstract.** This paper presents early results on applying causal inference to evaluate LLM-as-a-Judge for software maintainability. The causal graph reveals that certain research questions are not identifiable with our current setup and guides the design of future experiments. Our findings show that model size improves alignment with human experts only marginally, while increasing code size drastically reduces it.

**Keywords:** Software Engineering · Causal Inference · Software Maintainability

## 1 Introduction

Evaluating the quality of LLM-generated software artefacts remains challenging: human evaluation is costly and unscalable, while traditional metrics offer limited insight. *LLM-as-a-Judge* approaches provide a promising middle ground, yet suffer from unresolved issues around reliability, bias, and standardization [5, 12, 13]. This paper presents early results of the evaluation of an LLM-as-a-Judge approach on a maintainability task through the lens of causal inference. Beyond measuring alignment between Qwen-3.5 (in different sizes) and human experts, we use causal inference to identify which factors influence this alignment.

## 2 Related Work

Several studies have explored LLM-based evaluation across SE tasks (see also reviews [4, 5]). For **code generation, translation, and summarization**, Wang et al. [14] benchmarked seven LLM-as-a-judge methods on three SE datasets; Zhou et al. [16] proposed SE-Jury, an ensemble judge on diverse benchmarks; and Fandina et al. [2] introduced REFINE for benchmarking evaluators on industrial COBOL data. For **code annotation and quality assessment**, Ahmed et al. [1] tested six LLMs on ten tasks across five datasets, while Kumar et al. [7] evaluated GPT-4o, LLaMA 3.1, and Mistral Large on commit messages. Regarding **alignment and verification**, Weyssow et al. [15] built CodeUltraFeedback (10K instructions, 14 LLMs) to fine-tune CodeLlama-7B-Instruct, and Jin et al. [6] exposed systematic LLM failures in verifying code against specifications.

To the best of our knowledge, this paper is the first to employ causal inference for evaluating LLM-as-a-Judge.

## 3  Methods

### 3.1  Dataset

The evaluation dataset, collected by Schnappinger et al. [10, 11], contains 304 Java classes from 5 projects (ArgoUML, AOI, Diary Management, JUnit 4, JSweet). Each class is assessed on five maintainability aspects using a 4-point Likert scale (strongly agree to strongly disagree), where values represent expert answer probabilities aggregated from multiple experts: Readability (Rd, "this code is easy to read"), Understandability (Ud, "the semantic meaning of this code is clear"), Complexity (Cx, "this code is complex"), Modularity (Md, "this code should be broken down into smaller sections"), and Overall maintainability (Ov, "overall, this code is maintainable"). Note that Cx and Md are negatively worded: "strongly disagree" indicates maintainable code. The dataset is imbalanced; Table 1 details the distribution of dominant answers across all dimensions.

|                   | Ov. | Rd. | Ud. | Cx. | Md. |
|-------------------|-----|-----|-----|-----|-----|
| strongly agree    | 174 | 183 | 157 | 22  | 29  |
| weakly agree      | 64  | 79  | 76  | 41  | 31  |
| weakly disagree   | 41  | 38  | 51  | 60  | 49  |
| strongly disagree | 25  | 4   | 20  | 181 | 195 |
| total             | 304 | 304 | 304 | 304 | 304 |

**Table 1.** Number of time the probability a given answer (strongly agree, weakly agree, weakly disagree, strongly disagree) is maximum ($\geq 0.5$). Most of the time experts strongly agrees that the code provided is overall maintainable (Ov.), readable (Rd.), understandable (Ud.), and strongly disagree that the code is too complex (Cx.), and that is should be broken down (Md.).

### 3.2  LLM-as-a-judge Setup

**Hardware:** Experiments were run on an NVIDIA DGX B300 system[1] with 8×NVIDIA B300 288 GB GPUs (2304 GB total).

**Models and Servers:** We chose the following models (and model servers): qwen3.5-4b (Ollama)[2], qwen3.5-9b (Ollama)[3] both with default configuration ($top_k = 20$, $top_p = 0.95$ and temp. $= 1$), and Qwen3.5-397B-A17B4 (vLLM v0.21.0)[4] with default parameters ($top_k = 20$, $top_p = 0.95$ and temp. $= 0.6$).

**Prompt:** Figure 1 shows the prompt used for the judge model. Questions were the same as in the dataset (see previous section). The code snippets where the source code of each Java class.

---

[1] https://www.deltacomputer.com/nvidia-dgx-b300-2304gb.html
[2] https://ollama.com/library/qwen3.5:4b
[3] https://ollama.com/library/qwen3.5:9b
[4] https://huggingface.co/nvidia/Qwen3.5-397B-A17B-NVFP4

```
Here is some code:
{code}

Question: {question}

Please provide your answer and a brief explanation.
Answer with exactly one of these options: strongly agree, weakly
agree, weakly disagree, strongly disagree

Format your response as follows:
Answer: [your answer]
Explanation: [your explanation]
```

**Fig. 1.** Prompt used for the experiments.

### 3.3 Metrics

We measure agreement between the reference answer (i.e., the human-provided ground truth) and the LLM's answer using three categories: **Exact match:** Both answers are identical. **Partial match:** Both answers fall on the same side of the scale but differ in intensity (e.g., *strongly agree* vs. *weakly agree*). **Wrong direction:** The answers fall on opposite sides of the scale (e.g., one indicates agreement while the other indicates disagreement).

### 3.4 Research Questions and Identifiability

To identify which factors causally influence model accuracy, we adopt Pearl's structural causal inference framework [9]. We encode our assumptions as a directed acyclic graph (DAG, Figure 2), where nodes represent variables and edges encode direct causal effects. Using the back-door criterion, we determine valid adjustment sets for unbiased effect estimation and assess which research questions are answerable given our experimental design. We formulate three research questions regarding the causal effect of different factors on evaluation accuracy.

*RQ1: What is the causal effect of model size (MS) on accuracy?* This would tell us whether using bigger models is better for aligning the judge evaluation with the human ones. This question is **answerable** within the current setup. No back-door path exists; the unadjusted association directly estimates the causal effect.

*RQ2: What is the causal effect of the phrasing of the prompt on accuracy?* Here our setup allow us to investigate two aspects: the question type and the phrasing direction. Assessing the effect of the question type is answerable in our setup. There is no back-door path and the unadjusted association directly estimates the causal effect. For the phrasing direction, the question is **not answerable** with the current setup. The back-door criterion requires conditioning on Question Type (QT) to block confounding paths (PD←QT→GT→AC and PD←QT→MA→AC). However, in our design each question type has exactly

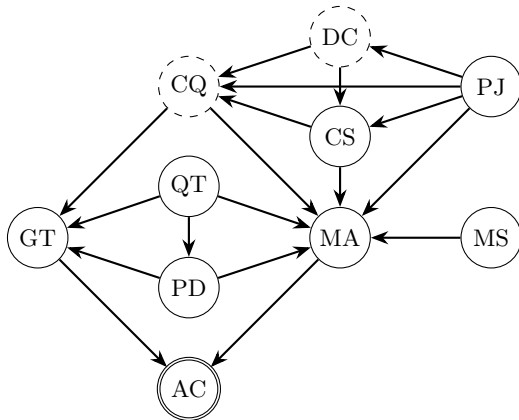

**Fig. 2.** Causal DAG. **PJ**: Project (AOI, ArgoUML...), **CS**: Code Size, **CQ**: Code Quality (latent), **DC**: Development Context (latent): captures within-project confounders (developer experience, code purpose, code age...), **QT**: Question Type (Ov., Rd., Ud., Cx., Md.), **PD**: Phrasing Direction (positive: agree means the code is maintainable as in Ov., Rd. and Uv, negative: disagree means the code is maintainable as in Cx. and Ms.), **MS**: Model Size (4B, 9B, 397B-A17B), **GT**: Ground Truth ([strongly|weakly] [agree|disagree]), **MA**: Model Answer ([strongly|weakly] [agree|disagree]), **AC**: Accuracy (outcome). Dashed border = latent variable; double border = outcome.

one phrasing direction; conditioning on QT therefore absorbs all variation in PD. Addressing this question would require a design in which each question is posed in both positive and negative form (e.g., "The code is readable" and "The code is not readable").

*RQ3: What is the causal effect of code characteristics on accuracy?* This question investigates whether the code under evaluation itself influences judge performance — e.g., LLM-as-a-Judge may align well with human experts on small code snippets but degrade on larger inputs. We operationalize this through **code size** (measured in input tokens), which has a direct mechanical relationship with LLM performance: larger inputs strain context capacity and attention mechanisms, providing a plausible causal pathway (CS→MA) independent of code quality. Two back-door paths run through project (PJ) (CS←PJ→MA→AC and CS←PJ→CQ→GT→AC); conditioning on PJ blocks both.

However, we acknowledge that this adjustment relies on the assumption that *within-project* variation in code size is not systematically driven by factors that also affect code quality. In practice, unobserved confounders (DC) — such as developer experience, code purpose, or code age — may induce residual confounding not blocked by Project alone. Results for RQ3 should therefore be interpreted as *suggestive of a causal relationship* rather than strictly causal.

## 4    Results

### 4.1    RQ1: Effect of model size (MS)

Figure 3 shows the impact of model size on LLM-as-a-Judge answers accuracy. Larger models yield more exact matches and fewer wrong-direction errors, indicating better alignment with human experts — consistent with prior findings [3, 8]. Assuming a linear relationship and no confounding from architecture (dense vs. mixture-of-experts) or serving infrastructure (Ollama vs. vLLM), each additional 1B active parameters is associated with a $+0.887$ pp increase in exact matches, $-0.298$ pp in partial matches, and $-0.590$ pp in wrong-direction errors.

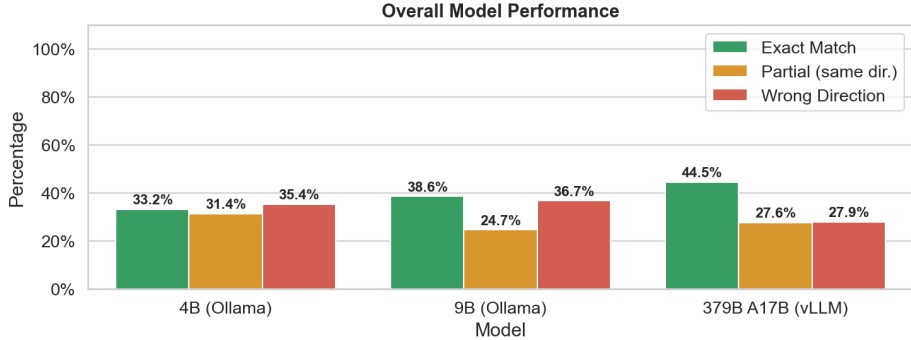

**Fig. 3.** Effect of the models on the quality of the answers of the LLM-as-a-Judge approach

### 4.2    RQ2: Effect of question type (QT)

Figure 4 shows the differences between the three models for different types of question. We can see that, in terms of readability and understandability, all models tend to be equally aligned, with slightly better alignment the bigger the model. However, for complexity and modularity, the larger model is significantly better aligned than the smaller ones. This may be because both tasks require stronger reasoning capabilities.

### 4.3    RQ3: Effect of code size (CS)

Our initial analysis reveals a nonlinear effect: model alignment decreases as code size increases and then plateaus. Due to space constraints, Figure  5 only shows this relationship for the largest model, but smaller models exhibit a similar pattern, albeit less smoothly.

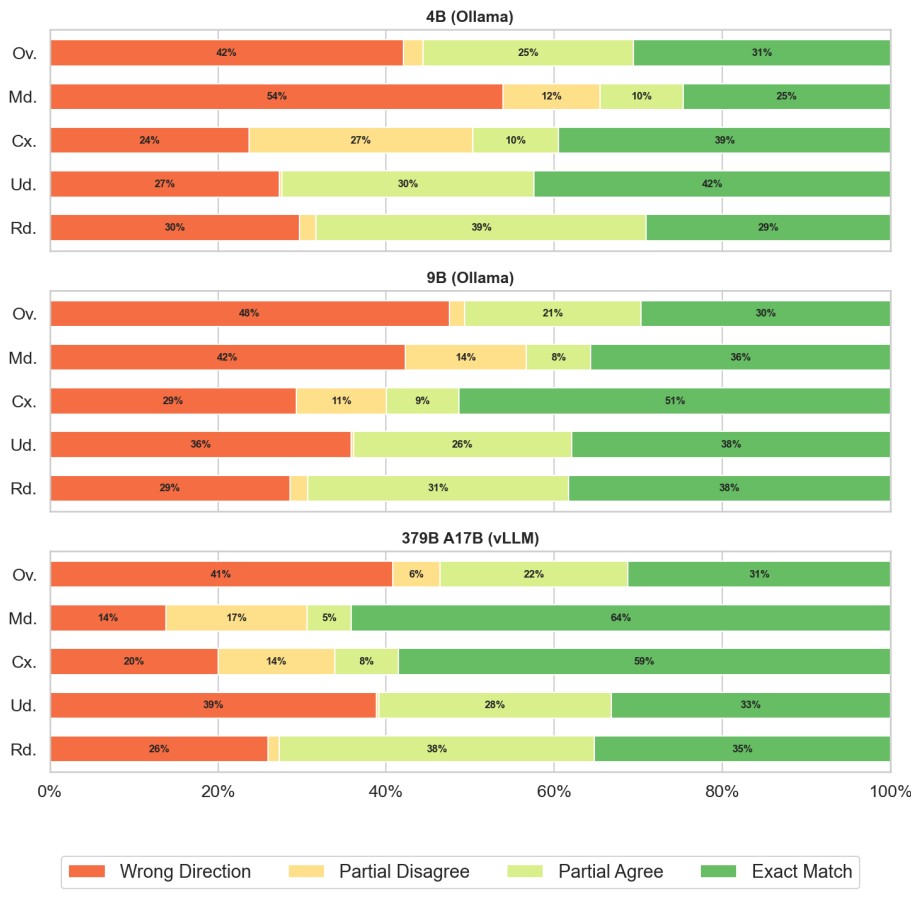

**Fig. 4.** Alignment metrics for models and question types.

## 5    Discussion and Conclusion

This paper presents early results of an LLM-as-a-Judge approach for estimating Java class maintainability. While many aspects remain to be consolidated (e.g., varying temperature, sampling strategies, model server effects, assessing bias left by uncontrolled confounders in RQ3), we offer one of the first attempts to use causal inference for assessing which factors influence the reliability of LLM-as-a-Judge. As such approaches become more prevalent, principled methods for empirically evaluating their reliability are needed. We hope this work provides directions for future research on this topic.

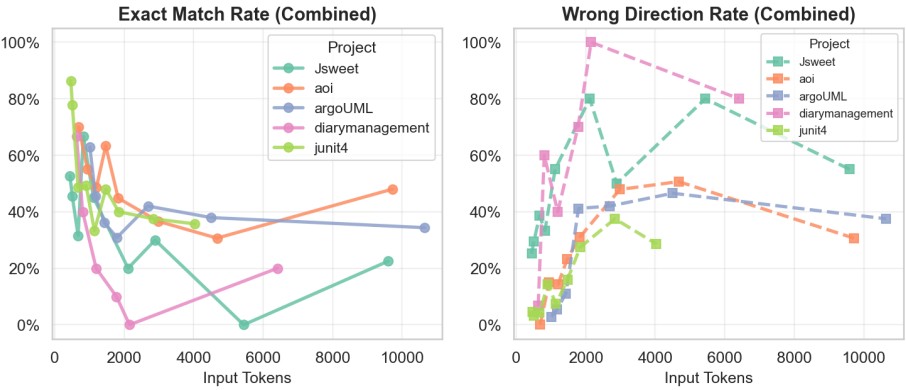

**Fig. 5.** Quality of the answer (exact matches (left) and wrong direction (right)) vs. code size (measured in number of tokens). Each point represent a decile.

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
