# OpenReview forum: "LLM-as-a-Judge for assessing maintainability. First results with a causal approach."
_KI/2026/Workshop/AI4SE — AI4SE Workshop_

### Official Review · Reviewer_Aswz · 2026-06-12
**Interesting research in progress with improvement potentials**

**Rating:** 6
**Confidence:** 4

**Review:**

Relevance of paper:
The paper discusses the merits of using LLM-as-a-Judge approaches for the evaluation of software code quality in the context of software maintenance. The author develops a causal model to use as a reference for guiding the evaluation of three research questions related to the effects of model size, phrasing and code size on outcome accuracy. The author then evaluates three (open source) LLMs of different sizes with regards to their performance. The paper shares the outcomes of this evaluation and briefly discusses them.
The topic is highly relevant for research and practice, as it addresses an important challenge with an increasing amount of code being generated, while we experience at the same time that human capacities become the bottleneck for reviews. To further strengthen the relevance of this (ongoing) research work it would be advisable to have a look into commercial SE-Copilots and evaluate the performance of their models as these are far more specific for SE tasks.

Structure and clarity of paper:
Overall, the paper is written well and follows a logical structure. From a structural perspective the sections that address related work could be improved (see also review suggestions below in section that addresses related work). Furthermore the section that addresses research methodology requires improvement and more clarity of the execution of the evaluation methodology (see dedicated review section).

Related work:
The paper addresses some of the relevant related work in the domain but unfortunately seems to have a focus on more general or open source LLMs. In the field of SE we have seen a number of commercial SE-specific coding assistants (e.g. GitHub Copilot). Prior to publication this paper should provide a perspective on those and discuss potential  limitations of the current more generalist approach. Discussion and outlook could reference this important domain, which is highly relevant in industrial practice and would raise the relevance of this work going forward.

Research methodology:
This paper unfortunately lacks a dedicated and concise description of the research methodology applied for designing and evaluating the artifact under study. Hence, before jumping into dataset and setup the paper should provide a short description of the overall research methodology and how it ties in the causal approach in order to improve transparency wrt. the research approach across the phases of design and evaluation.
The paper sates several times that it addresses causal inference but actually only provides a qualitative causal model with some descriptive comparisons of measurements. In its current form it lacks the statistical robustness and hence it would be advisable to tone down claims around causal inference statements. Similarly statements like “each additional 1B active parameters is associated with a +0.887pp increase in exact matches, −0.298pp in partial matches, and −0.590pp in wrong-direction errors.” create the perception of quantitative precision that cannot be obtained and should not be claimed based on the current scope of research. Please rephrase to indicate trends/directions that can be derived from your research.

---

### Official Review · Reviewer_ceps · 2026-06-12
**A novel application of causal inference to LLM-as-a-Judge evaluation for software maintainability, with clean experiments and actionable findings**

**Rating:** 8
**Confidence:** 4

**Review:**

#### Summary
The paper applies Pearl's structural causal inference framework to evaluate LLM-as-a-Judge for software maintainability. Beyond measuring alignment between LLMs (Qwen-3.5 in 3 sizes) and human experts, the authors employ causal inference to identify which factors influence this alignment.
Using an existing dataset of 304 Java classes with expert annotations across 5 maintainability dimensions, the authors formulate 3 research questions concerning the causal effects of model size, prompt characteristics, and code size on judging accuracy. A structural causal model (DAG) is used to reason about identifiability and adjustment sets, and to assess which research questions are answerable given the experimental design. The results suggest that larger models provide only modest improvements in alignment with human judgments, whereas increasing code size substantially reduces agreement.


#### Strengths
- The methodological perspective to apply causal inference to evaluate LLM-as-a-Judge reliability is novel and also interesting for a broader range of empirical LLM studies. The identifiability analysis is principled and valuable, the paper explicitly explains which research questions are and are not answerable with the current experimental design.
- Assessing code maintainability is a highly relevant problem. The experiments are straightforward and reproducible, with a clearly described prompt, and the use of Schnappinger et al.'s multi-expert maintainability dataset provides a credible human-annotated ground truth instead of relying on synthetic labels.
- The practical findings that larger models only marginally improve alignment while performance degrades substantially with code size are actionable.

#### Weaknesses
- More details on the causal estimation procedure would improve reproducibility. The paper discusses identifiability but provides relatively little information on how effect estimates are actually computed.
- For RQ1, i think it is a strong assumption that different model architecture, inference servers or decoding parameter do not influence accuracy (in this otherwise methodological sound study)
- The dataset consists of only 304 samples, it would strengthen the study to report confidence intervals or significance tests
#### Detailed comments for author:
- small typos:
	- 2 -> (see also reviews [4, 5])
	- 3.2 The code snippets w~~h~~ere the source code of each Java class.
- For RQ2, it would be interesting to see the experiment using positive and negative phrasing, are you planning this evaluation?
- Why is a linear relation assumed in RQ1, Figure 3 does not appear to be linear?
- Adding the missing models to Figure 5 (perhaps in supplemental material) would be valuable
- The metrics in Figure 4 do not match the 3 categories from Section 3.3